# Paradigm Lost

**DOI:** 10.3390/cancers17132187

**Published:** 2025-06-28

**Authors:** Jane Mellor, Ewan Hunter, Alexandre Akoulitchev

**Affiliations:** 1Department of Biochemistry, University of Oxford, Oxford OX1 3QU, UK; jane.mellor@bioch.ox.ac.uk; 2Oxford BioDynamics Plc, Oxford OX4 2WB, UK; ewan.hunter@oxfordbiodynamics.com

**Keywords:** 3D genome architecture, capturing chromosome conformation (3C), epigenetics, metabolism, phenotype, clinical diagnostics

## Abstract

**Simple Summary:**

Clinical outcomes can be linked to the fundamental role played by the regulatory 3D architecture of the genome in a living cell and the unique properties offered as a molecular biomarker modality. Here, we describe how this molecular phenomenon acts as an imprint of the cellular genetic, epigenetic and metabolic status which form discriminatory stable conditional structures that inform on the state of the cell and its phenotype. Consequently, 3D architecture, as a biomarker, offers a more comprehensive translation from molecular readouts to clinical outcomes, with examples and evidence provided by the highly accurate diagnostic and prognostic clinical tests already reduced to practice in the US and UK. Moreover, with support from Large Language Models, we can elucidate the complexities and inter-relationships of molecular pathways and regulation using a 3D genomic architecture knowledge base, to gain insights into various molecular modalities and associated clinical conditions.

**Abstract:**

**Background/Objectives**: The 3-dimensional (3D) architecture of the genome in the nucleus of a living cell plays an unexpected yet fundamental regulatory role in cell biology. As an imprint of the cellular genetic, epigenetic and metabolic status, it discriminates pathological conditions through conditional changes to long-range 3D interactions (up to 300 kb) and thus could act as a powerful molecular biomarker linked closely to clinical outcomes. **Methods**: Here an assessment is made of the latest paradigm shift in molecular biology from a supply chain where information flows from DNA to RNA to protein, to the concept of heritable 3D folding of the genome reflecting the epigenetic and metabolic state of the cell, and which serves as a molecular biomarker for complex clinical outcome. **Results**: While biomarkers based on individual components of the supply chain fail to accurately reflect clinical outcomes, 3D genomics offers highly informative insights, exemplified for immuno-oncology and prostate cancer diagnosis by clinical tests of superior performance, already in practice in the US and UK. **Conclusions**: A more complete understanding of human biology will require models that account for the flow of information to and from the 3D genomic architecture in living cells, together with its regulation and logic. Integrating these principles into biomarker discovery and therapeutic design, along with other frontline approaches in precision medicine, including multi-omics and other system-level tools, will be essential for advancing precision medicine beyond its current limitations.

“What is dark within me, illumine.”—John Milton, Paradise Lost

## 1. Biomarkers View

What has prompted a recent flow of forceful commentaries with such brave titles as ‘The end of the genetic paradigm of cancer’ [1] or ‘The cellular dogma’ [2], to mention just a few? What has united life science researchers from Tufts University, The System Biology Institute, Ecole Normal Superior, The Chan Zukerberg Initiative, or The University of Oxford in raising their voices against established paradigms in the field of fundamental biology?

Is it true, to quote some, that today’s genetic research has swapped ‘deep thinking for deep sequencing’ [1], with the science largely being reduced to a transactional operation of cataloguing molecular alterations and looking for associations [1]?

The state of the union between academic life sciences, the health industry, and societal expectations is much worse than that. Academic life sciences, armed with the fundamental insights into molecular biology, have been promising us exciting breakthroughs in disease treatment for several decades now. Does one remember claims that the human genome project will become a book of life for disease treatments? That cancer would be conquered in 20 years. If only more funding is given to the academic research community, then the breakthroughs in healthcare, powered by science, will be achieved at a cascading speed. Understandably, any subject related to health and disease is a legitimate topic of discussion, with wide-ranging opinions held. Unlike most highly technical science fields, life science holds us in a psychological and emotional grip. We want new health breakthroughs. We feel much more entitled to judge and expect more. Today, much of the credibility of science is tied up in its promises to solve current health challenges.

And here comes the first lesson in history.

### 1.1. First History Lesson

One of the most famous stories about how science translated itself into a great power was Robert Oppenheimer’s nuclear bomb “Manhattan project”. When the pilot plutonium bomb nicknamed “Gadget” exploded, everyone understood that the incomprehensible science of nuclear physics worked. Science proved to be powerful, even if it was hard to follow for the non-specialist. In fact, the success of the Manhattan project convinced the US Federal Government to start the systematic funding of science, expanding eventually to molecular biology and life sciences. Today, science occupies the central stage in the life of our society especially when related to human health and combating illnesses. It is a powerful mix, both economic and moral, and backed by some of the largest government expenses for health and social care. Despite remarkable progress in vaccine development driven by the COVID-19 pandemic, saving millions of lives, many expressed their dissatisfaction with the experts involved, wishing for better outcomes. Our doctors, just like their predecessors, still face a very challenging world of complex and often confusing patient cases, counterindications, diagnostic and prognostic uncertainties, changes in responses to treatment, relapses and side-effects, yet society expects more from health care and life sciences. This jaundiced view of experts and life science extends to its commercialization. A cruel joke among biotech analysts is that ‘most inventions in biotech turn out not to work, but a lot of people still manage to make money out of it’. And what happens when a biotech technology does work? Such a scenario is beautifully illustrated by a British innovation, the Humira treatment for rheumatoid arthritis. It was sold off quickly and cheaply. Note to the reader, Humira (D2E7)) was developed using the phage display technology from the Laboratory of Molecular Biology (Cambridge, UK), followed by joint efforts from Cambridge Antibody Technology and BASF. In 2022, when Humira ended up in the hands of AbbVie, it generated record revenues of over USD 21 billion.

And here comes the second lesson in history.

### 1.2. Second History Lesson

Over the last 20 years, doctors and pharma have witnessed poor outcomes in the clinic, ranging from high numbers of false positives and false negatives in patient diagnosis to significant numbers of patients not responding to the drug therapies. If you are a man over 50, and your basic PSA test for prostate cancer is over the threshold of 3 ng/mL, it will turn out to be a false call in three cases out of four [3]. If you are a renal cancer patient and your oncologist is considering a powerful immune oncology treatment that could save you, the chance that you will respond to the treatment is around 26%. If you do not respond, you are still likely to develop immune-related adverse events (70%), i.e., autoimmune side effects, or, in an even worse scenario, turn out to be a hyper progressor (circa 11%), which is when the drug will speed up growth of the cancer. As a predictor of a patient’s response in advance of immunotherapy treatment, the standard IHC PD-L1 test (Agilent Technologies, CA, USA), based on protein levels of expression in the biopsy of the tumour, will deliver its result with an average accuracy of 40% [4]. Finally, the average failure rate in drug development is over 90% [5], despite claims that the mechanism of action of the therapeutic agent is understood.

Why are we failing? The complexity of biological systems confounds our optimism in the power of science to generate simple solutions, our strict adherence to a few molecular biology paradigms, our systematic molecular cataloguing of data, our reductionist logic and our enthusiasm for AI to generate answers. The fundamental designs of biology do not tolerate oversimplistic reductionist interpretations and simplistic mechanisms. This leaves us repeatedly challenged by the uncomfortable evidence of failures (more about this later). As a scientist, it prompts one to look for new interpretations and new paradigms, as a correction to the assumed scientific fundamentals. The choice is between losing faith and defaulting on promised solutions or re-evaluating our insights into biological fundamentals.

### 1.3. Paradigm Lost

The Nobel Prize of 1965 was awarded to Francois Jacob, Jacque Monod and Andre Lwoff for their work establishing a key role for RNA in the flow of genetic information from DNA to protein synthesis. The famous battle cry of molecular biologists: ‘DNA makes RNA makes protein’ has become the backbone of molecular biology, with the key cellular ‘mechanisms’ focused on this information flow [2]. The flow has become the Central Dogma of molecular biology. Not surprisingly, it has attracted the attention of biomarker developers, trying to find molecular readouts that would guide us on what is happening clinically to the patient. In an obvious way, DNA, RNA and proteins offer a rich milieu of biomarker readouts, from genetic risks to transcriptional and protein profiles. And here comes the clinical reality—such an approach delivers high false-positive and high false-negative error rates for real patients in trials and clinical practice.

For example, diagnostic and prognostic biomarkers built on *genetic risks* can serve only a few single-factor genetic diseases. Most clinical conditions are multi-factorial. Genetic risk factors, alongside other inherent complexities, necessitate impractically large effect sizes within development cohorts, largely due to the low prevalence of *pathogenic mutations*. The development of a robust genetic biomarker may require the recruitment of upwards of 10,000 affected individuals. Moreover, even when such mutations are present, they often demonstrate incomplete penetrance, failing to consistently manifest as overt pathological conditions in individuals. Importantly, the low prevalence of causal variants generally correlates with small individual effect sizes, underlining the fact that most complex diseases are not attributable to single gene defects, but rather involve multi-loci interactions. This points to underlying network effects driving disease phenotypes, challenging the prevailing molecular dogma that posits single-target mechanisms as the primary basis for pathology.

*Gene expression* is a long-favoured target for biomarker development, representing direct expression from the genes often genetically linked to the condition. Monitoring gene expression also offers a more specific output, as diseases arise in the context of individual tissues, while the genetic risk is represented in all cells in the body. This also means the focus can be on the 24,000 or so genes encoding mRNAs and proteins. However, the genome is much more complex, and it is not just the genes that are transcribed, but practically the whole genome on both strands of the chromosomal DNA [6]. And, to make it even worse, around 90% of the messenger RNA produced from the genes may not leave the nucleus, and so never reaches the sites of the protein synthesis [7,8]. One of the consequences of this is that profiles of protein levels and RNA levels often demonstrate poor concordance [9,10].

This is not meant to be a criticism of the way nature has designed the cell. What has been described here is a beautiful, effective, robust and flexible supply chain process (Figure 1): one which can accommodate fast changes in the amplitude of supply and sustain full capacity for production of proteins, thereby meeting the need for fast or slow kinetics. However, such a supply chain is not good at providing a judgment about the exact phenotype of the cell at a given point. Trying to predict phenotype from the supply chain is no different from trying to figure out how many Samsung mobile phones will be bought in the UK this year, based on overall number of microchips made by the Taiwan Semiconductor Manufacturing Company last year. Molecular biomarkers based on supply chain components will be *reflective* but inaccurate tools for predicting clinical outcomes. High level of *false positives* and *false negatives* can be attributed to variations within the supply chain. Even protein production, providing enzymes and building blocks of the cell, is not the final step in the story of a supply chain. Current understanding acknowledges post-translational modifications, which are often tied to the metabolic state of a cell, as well as extensive interactions between protein, DNA or RNA, as being critical switches that determine the final biological activity for many of these proteins and, in turn, have a much greater impact on final phenotype.

Meanwhile, our view of DNA as a store of heritable genetic information has changed, as it is now appreciated that the way the information in DNA is used by a cell or organism can be moderated *epigenetically*. DNA itself can be modified by adding methylation to certain bases. The DNA also exists as chromatin, a polymer of DNA and histone proteins, which carry an addition code comprising more than 50 different post-translational modifications, together known as *epigenetic modifications*. Interestingly, many of these epigenetic modifications are maintained by enzymes closely linked to metabolic controls, so we might take a pause and think of it provocatively as *metabolic* rather than epigenetic control. The most fascinating development in understanding chromatin is how ≈1.8 m of DNA is packed into each cell nucleus with a diameter of ≈10 microns. We now recognise that this packaging is non-random, it has strong regulatory and heritable features, and that these are reproduced upon division in the daughter cells [11,12]. A vast body of work on understanding how chromatin folds, often described as the 3-dimensional (3D) genome, reveals that this folding acts as an integrator of the many various molecular regulatory mechanisms and events that contribute to cellular phenotypes [13]. If it can be captured, this folding introduces a functional readout for the cellular phenotype and thus can be used as a biomarker. Three-dimensional folding helps to reduce the impact of stochastic noise and uncertainties about where to set significance over background and collapses the complexity of the vast molecular datasets. Every time one sees a specific phenotype in a cell or organism, there is a very consistent conditional 3D genomic fold which acts as a molecular footprint of that phenotype.

Mathematics has profound insights into how we understand the topological puzzle of 3D folding of the genome. A polymer folding into limited space will be defined and distinguished in topology by distant sites coming into close juxtaposition. Biology will describe the same phenomenon as an active genome in a living cell forming preferential and conditional long-range interactions, a physical manifestation of genomic regulatory networks of mutual influence in a cell. Long-range interactions create closed topological domains, establish conditional regulatory borders across the genome, and act as the scaffolding template for the massive molecular machinery of gene expression, division, and repair, to build and maintain the cell. The 3D genomic profile of the cell is a stable, heritable feature that uniquely reflects its genetic, epigenetic and metabolic state, the integrated imprint of the cellular microenvironment, organismal systems dynamics, and external environmental exposures. In addition, the 3D genomic profile is stable and reflects the phenotype (or clinical condition), as long as it exists. Such a stable biomarker is a great advantage, as you do not have to monitor volatile metabolites in conditions such as Metabolic Dysfunction-Associated Steatohepatitis (MASH) and Type 2 Diabetes Miletus (T2DM), or to distinguish between fast and slow transcriptional activation that defines polarization of naïve T cell in autoimmune conditions [14].

### 1.4. Good Biomarkers?

A word of caution, given the idea that the 3D genome integrates a wide range of molecular inputs; can we not use a systems biology approach leveraging machine learning (AI) rather than assessing 3D genomic folding directly? The answer seems to be no; these sorts of analyses revealed that the integration of molecular features from genetic, transcriptional, and epigenetic datasets, even within large oncology cohorts, offer limited improvement in patient stratification relative to clinical assessments and outcomes [13]. But another key question is whether direct assessment of the 3D genomic folding will turn out to be yet another overpromising and overhyped biomarker modality. The answer to this is also no; they are not overhyped, and evidence for this lies within the real world of clinical practice.

When assessment of the 3D genomic folding is used with a biomarker platform which supports quality control to industry standards (ISO 13485 [15], ISO 9001 [16] and ISO 15189 [17]), and is combined with independent validations, involving accredited clinical labs in the US and the UK, these 3D genomic biomarkers have delivered robust clinical tests, which overcome the long-standing challenges in patient stratification. It is important to mention that the method developed to capture and detect 3D interactions in chromatin, previously described in [3,4], benefits from reproducible high solubilization of large chromatin targets and use of functional filters in detection, resulting in a significant reduction in stochastic noise, so prevalent in Hi-C and other related common protocols. Regarding data analysis and deployment of the appropriate AI tools, Large Language Models (LLMs) have been revolutionizing biomedical research by connecting vast corpora of textual and structured data. However, LLMs that fail to incorporate genome conformation data are fundamentally incomplete. Missing this spatial regulatory layer can lead to misinterpretation of causal relationships, poor biological inference, and inaccurate predictions in translational and clinical contexts. To address this, we integrate BERN-derived biomedical knowledge with EpiSwitch^®^ 3D predictions and empirical EpiSwitch 3D interaction data using graph neural networks, enabling more biologically complete and accurate inferences. Incorporating genome conformation is essential for any next-generation AI platform aiming to decode complex disease biology.

Immune oncology offers treatment with immune check-point inhibitors (ICI) that could lead to remarkable and enduring benefits to patients [4]. However, the objective response rate to ICI is about 20%, on average, across many cancer indications. When this is combined with the high cost of the treatments and the high level (70%) of auto-immune side effects (immune-related adverse events), identifying potential responders in advance is quickly becoming an urgent requirement in the clinic. The EpiSwitch^®^ Checkpoint Inhibitor Response Test (CiRT) (Oxford BioDynamics, Oxford, UK) is based on the detection of eight 3D genomic markers in patients’ blood and delivers 84% accuracy for a range of cancers [4]. CiRT has its own unique US reimbursement code, has NY State CLEP clinical approval, and is approved for use in the UK by BUPA and AXA Health. The US registry trial PROWESS is exploring longitudinal data specific to the clinical utility of CiRT and its place in the US treatment guidelines. Blood samples for analysis are shipped at ambient temperature to the CLIA (US) or UKAS (UK) accredited clinical labs, thanks to the biomarkers’ high stability. Sample turnaround is 4–5 days upon receipt. With 84% accuracy, CiRT out-performs the current tools for predicting the response to ICI. These include a biopsy-based immunohistochemistry (IHC) test for expression level for the PD-L1 protein with an overall accuracy of around 40% [4] and tumour burden biopsy-based sequencing markers, with notably poor meta-data accuracy [4,18].

Another example concerns prostate cancer. There is no national screening programme for prostate cancer in the UK, reflecting the poor state of the common clinical biomarkers. Current reliance on Prostate-Specific Antigen (PSA) testing is undermined by its poor specificity (53%) and sensitivity (64%), raising concerns about overdiagnosis, overtreatment, and uncertain mortality benefits [3]. Prostate cancer remains the leading cause of cancer-related death among men over 50 in the UK, yet no effective screening or risk-stratification strategy has been implemented. Although PSA testing offers reasonable negative predictive value (86%), with low levels reliably indicating absence of disease, its positive predictive value remains poor (25%). Most men with PSA levels above 3 ng/mL do not ultimately have clinically significant prostate cancer. Yet, elevated PSA typically triggers costly and invasive diagnostic pathways. Biopsy remains the gold standard for diagnosis, yet, without precise lesion targeting, its sensitivity is also limited, compounding the challenges in accurately identifying clinically significant disease. Magnetic Resonance Imaging (MRI) can detect abnormalities within the prostate, including inflammation and malignancy, but lacks the specificity to distinguish between them, yielding a positive predictive value of only 51% [3]. Moreover, NHS patients in the UK often face significant delays in MRI access.

The 3D genomic platform has been used to develop the non-invasive diagnostic EpiSwitch^®®^ Prostate Cancer Detection Test (PSE) (Oxford BioDynamics, Oxford, UK), a blood-based test aimed at the early evaluation of men for the presence of prostate cancer. The PSE test has two components: an individual’s PSA readout, with its helpful negative predictive value (NPV), and five systemic 3D genomic markers which, when combined, deliver a test of 94% accuracy [3]. Currently, the test is covered by BUPA UK and available for private health patients in the UK (for example, the Goodbody Clinic, The London Clinic), Romania and Turkey. In the US it also has a unique reimbursement code and has accumulated real-world evidence for its clinical utility. Patients and their doctors, from 48 of the 50 US states, have benefited from PSE. According to Dr G. Polman, who runs the national Men’s Health podcast, his clinic has avoided unnecessary biopsies in over 50% of his patients, thanks to the early use of PSE. In the UK, Prostate Cancer Research, one of the leading charities, has strongly advocated for the use of PSE among patients at high risk. Their charity ambassador Hon Rishi Sunak, the former UK Prime Minister, has added his voice to the Government discussions, advocating routine testing for high-risk men. NHS patients are still waiting for access to PSE.

### 1.5. Can You See This in Blood?

A key advantage of 3D genomic profiling lies in its potential as a liquid biomarker with signatures present in white blood cells (WBCs). How can the definitive characteristics of a cancer in one part of the body, or a neurodegenerative disease in the brain, be present in a patient’s blood? It has long been known that there is active crosstalk between cells in a multi-cellular organism, often detectable as changes to the epigenome. For example, characteristic DNA methylation footprints in schizophrenic patients are evident in their blood [19]. We now have a much clearer understanding of the systemic flow of molecular information around the body in exosomes, or extracellular vesicles (EVs). This leads to changes in WBC, including cells of adaptive and innate immune systems, as demonstrated experimentally for prostate cancer cells and in Cornell University studies [20,21,22] (Figure 2). Exosomes, as extracellular vesicles (EV), are highly abundant (10 × 10^9^/mL) and highly dynamic in blood, trafficking both metabolites and non-coding regulatory RNAs from the cells of origin throughout the body. Metabolites and non-coding RNA are powerful epigenetic regulators capable of modulating the epigenome of the recipient blood cells. Remarkably, in one day our blood is estimated to travel over 12,000 miles through our body. Thus, blood cells are a perfect tissue for 3D genomic readout of the footprints from local sites of any abnormality, even from the other side of the brain–blood barrier. Immune-cell surveillance of disease sites can also allow direct transfer of information reflecting the disease state in the blood cells [6,23].

In conclusion, the supply chain with its flow of information from DNA to RNA to protein was the focus of biomarker development, based on our assumption that this paradigm was all that was required to reflect the phenotype of a disease or cancer. However, repeated failures in the development of effective and informative biomarkers require a sea change in our thinking about how cellular phenotype is determined. Now, with 3D genomic biomarkers showing strong performance, and with the insights into the regulatory role of 3D genomics, the additional flows of information are waiting to be elucidated. Better insights into biology will have a better chance of delivering on its promises to understand and eventually cure complex diseases.

## 2. Biochemical View

Because current thinking in molecular biology is reductionist, genomes are often viewed not as a whole, but as a collection of individual loci on which molecular machines act to reproduce genetic information during cell division, DNA repair or gene expression. Although this provides helpful landmarks, it has turned out *not* to be a constructive way of thinking about how the phenotype of a cell or organism is determined, because we are far more than the sum of our genes. Genes and their expression contribute, but gene expression is a poor predictor of phenotype [9,10]. The gene-centric view is encompassed in the linear flow of information underpinning the central dogma (Figure 1 and Figure 3). Moving this focus to the nuclear architecture proves to be a more productive way of thinking about this problem. The nucleus is a dense and complex nano-environment, which partitions chemical reactions required to form chromatin and the 3-dimensional (3D) genome in the form of structural hierarchies. This packaging is fractal, a repeating unit of loops; thus, a change in the fractal dimension of the chromatin in one region will automatically lead to changes throughout the genome as the network reorganises itself, with changes in regions of compaction and accessible open chromatin, and thus 3D interactions [24]. Genomes are networks of local interactions determined by their local environment, which change in disease, during stress, and during biological rhythms (Figure 3). This environment comprises many highly charged molecules, such as the negatively charged nucleic acids, including DNA and long and short non-coding RNAs, and their positively charged interacting proteins that function to neutralise this charge and thus facilitate local packing and condensation. In addition, proteins associated with nucleic acids often contain inherently disordered regions (IDRs), which are the regions of proteins whose structure is fluid and hard to capture. Many DNA- and RNA-associated proteins contain IDRs, contributing to the formation of sub-structures within the nucleus at varying scales [25]. Other metabolites such as lipids, in nuclear lipid droplets, should also be considered in the dynamics and fluid organisation of these sub-structures [26]. Large sub-structures include the nucleolus, the best known membrane-less organelle or condensate, and the A/B compartments into which differentially expressed regions of different chromosomes are found, but many other sub-structures exist [27,28,29]. Some of these sub-structures involve poorly characterised genomic regions, but others such as the topologically associated domains (TADs) are well characterised. TADs often rely on proteins such as the CCCTC-binding factor (CTCF), whose orientation is proposed to determine the extent of ATP-dependent loop extrusion from the underlying compartments by cohesin, the ring-shaped ATPase [29]. TADs are envisaged to facilitate long-range (≈Mb) interactions in interphase chromosomes and are determined during differentiation from stem cells, being characteristic of a tissue type. Generally, unless a chromosome undergoes large-scale genetic rearrangements, as can happen in cancers, the TAD morphology is a consistent feature of the chromatin and associated with highly distinctive TAD boundaries [30]. Occasionally, disease-associated single nucleotide polymorphisms (SNPs) alter individual TADs, by changing the properties of boundaries; however, many more SNPs show no clear association with genes or TADs, but can be associated with some less-well-characterised 3D interactions that are *smaller* than TADs. This suggests that there is much more information in the 3D genome than currently appreciated, especially in the sub-structures associated with the poorly characterised genomic regions without obvious features, which often make the most informative biomarkers.

A good starting point to rationalise these ideas is to take a step back and explore factors that determine the biophysical properties of a local chromatin environment. These are envisaged to rely on the charges, interactions and structures formed between chromatin components. In addition to DNA–histone interactions, these include long non-coding RNA transcripts, non-coding enhancer RNAs (eRNAs), non-coding promoter divergent antisense transcripts (PDATs), the IDRs of transcription factors (often known as activation or repression domains), and the IDRs of histone proteins (the unstructured N-terminal histone ‘tails’). These interactions are thought to be another mechanism for facilitating many of the 3D interactions observed throughout the genome, including promoter–enhancer interactions and many interactions at regions whose function is not yet understood. Which interactions form are likely to change as the biophysical properties of the local environment change, for example, in diseases or stresses, where changes to the local concentrations of metabolites and ions, such as glucose, acetyl-CoA, alpha-ketoglutarate, SAM, lipids, NAD^+^, Fe^2+^, H^+^, Zn^2+^, K^+^ and H_2_O occur [31,32,33,34]. Subnuclear changes to metabolite and ion concentrations influence local covalent modifications in chromatin (as co-factors for nuclear enzymes mediating modifications to DNA, RNA and proteins), and to local ion and metabolite-dependent folds in proteins and nucleic acids, thus contributing to the networks of charges and interactions in the chromatin [35,36]. Although much of current thinking encourages a focus on the 25% of the genome involved in encoding genes and their regulatory elements, human genomes are subject to pervasive transcription which can alter chromatin modifications and create pools of non-coding RNA. These also participate in regional interactions and folding, determined by metabolites and ions, and so reflecting the metabolic and physiological state of a cell or tissue. As altered metabolism underlies a wide range of disease states, the structures adopted by the chromatin in cells will reflect these states.

Predicting where these interactions will occur and having a robust method to detect them is key to unlocking this new level of information in our genomes. Remarkably, machine learning algorithms can predict *potential* interaction sites, known as loop anchors. Millions of these are scattered throughout genomes, but extensive iterative experiments are required to find informative biomarkers that *link changes in the 3D genome to changes in phenotype*. Using capturing chromosome conformation (3C) technology, coupled with microarrays displaying the oligonucleotide fragments containing the two halves of potentially informative interactions, usually16 kb to 300 kb between interactions in cis, focused around the 4 bp cut restriction enzyme sites used to digest the crossed linked chromatin, two conditions can be compared and the small number of differential interactions that define each state uncovered from the millions of interactions that either do not change or are not informative enough to use to stratify ethnically diverse populations. Such technology, based on EpiSwitch clinical analysis of blood, coupled with AI-based EpiSwitch Data Knowledge [37] (Oxford BioDynamics, Oxford, UK) has been used to develop tests, now used in the clinic, to stratify individual patients in immune oncology for their responsiveness to immune checkpoint inhibitors (Checkpoint inhibitor Response Test: CiRT) [4] or to diagnose the presence of prostate cancer (PSE test) [3]. Of particular interest to basic scientists is gaining an understanding about how the unbiased integration of metabolic, genetic, epigenetic (epitype), gene expression (coding, and importantly, the long > 200 nt and short non-coding transcripts) information is reflected in the chromosome conformation signature (CCS) that precisely informs that patient’s disease phenotype and/or response to treatment. Interestingly, the loop anchors at the ends of these informative interactions sometimes have distinctive features, including disease-associated SNPs, open chromatin, regions enriched with transcription factor binding sites and local epigenetic modifications such as H3K27ac, which may be found within, or close to, genes associated with a disease. However, other anchors are found in regions without any obvious features, at least in terms of our current knowledge of the genome/epigenome.

How does this view of our genome relate to the central dogma? A widely held assumption is that changing levels of an mRNA transcript will “translate” into varying levels of a protein factor, which defines the phenotype, despite evidence to the contrary [9,10]. Furthermore, cells are programmed to maintain protein and osmotic homeostasis against perturbation from stress, disease and natural rhythms, conditions that are defined by large-scale variations in transcript levels [32,38,39]. Instead, these cells show changes in metabolites and post-translational modifications to proteins, allowing cells to adapt to their altered conditions, without large-scale remodelling of their proteome [39]. Remarkably, these events are also present in anucleate cells and cells lacking key transcriptional regulators, reinforcing the disconnect between transcript levels and protein levels [40,41]. So how can these observations be rationalised with the central dogma and the supply chain idea? What brings about changes in transcription and transcripts while protein levels remain constant? Interestingly, protein turnover by the proteosome, removing damaged or oxidized proteins, often occurs contemporaneously with new protein synthesis, to maintain the constant proteome [39]. This will require translation of the mRNA to produce a pulse of protein to replace that lost by turnover [32]. An interesting idea is that production of such a transcript is coordinated with protein turnover through protein activity in both the nucleus and the cytoplasm, which in turn reflect the metabolic state of a cell; for example, acetylation of histone proteins and transcription factors in the nucleus-promoting transcription and, at the same time, acetylation influencing the activity/targeting of the proteosome towards specific (damaged) proteins in the cytoplasm. This can lead to a perceived link between transcription and the phenotype of a cell, but may simply *reflect* altered metabolites associated with stress, disease or natural rhythms. However, as metabolites influence many more aspects of cell biology than the production of mRNA transcripts, assessing mRNA transcripts generally fall short as an accurate or comprehensive readout of phenotype. Generating metabolomes is challenging, especially as a diagnostic tool, so a proxy is required: the 3D folding of the genome.

## 3. Data Analysis View

Building predictive models from genetic and gene-expression datasets has historically been an ungratifying endeavour. Often, the realities of clinical outcomes defeated modelled expectations, exposing the gaps between molecular readouts and system-level behaviour. Despite advances in computational power and the rise of artificial intelligence, the fundamental barriers have remained largely unchanged, even as the promises have multiplied [42].

A critical question arises: is the data input adequate for the calculations we prescribe? Gödel’s Incompleteness Theorems [43] warn that in any sufficiently complex formal system, certain truths will remain unprovable from within the system itself. By analogy, if the input data and measurement languages we rely on in molecular biology are themselves too limited or misaligned with the true complexity of biological regulation, no amount of computational power, however sophisticated, can bridge that informational gap. Roger Penrose, awarded the Nobel Prize in 2020, has argued forcefully that consciousness, and, by extension, true intelligence, lies beyond algorithmic computation [44]. In this view, what we call “artificial intelligence” is better understood as advanced calculation, not true insight, a distinction with profound implications for the modelling of living systems.

This raises an urgent reconsideration of our current data analysis approaches to biological networks and self-organizing systems, whose adaptive behaviours often outpace the incremental timescales of Darwinian evolution. Can reductionist strategies, which deconstruct systems into isolated molecular parts, meaningfully capture the emergent properties of network regulation? Increasingly, evidence suggests they cannot [45,46,47].

In molecular biology, the persistent challenge of poor predictive biomarkers can be understood through the lens of foundational concepts from physics and complex systems theory (Figure 4). The El Farol Bar problem illustrates how reliance on partial, anticipatory information generates systemic unpredictability: agents make decisions without full system visibility, resulting in unstable outcomes [48,49]. Similarly, proxy measurements in molecular biology, single biomarkers, isolated pathways, or fragmented omic signals, are inherently insufficient to infer the dynamics of multi-layered biological networks, leading to instability in biological interpretations and irreproducibility in predictive models.

This measurement problem parallels the paradox of Maxwell’s Demon, which seemed to suggest that perfect information acquisition could reduce entropy. The resolution of this paradox underscores the fact that information itself carries physical constraints and costs. In molecular biology, the assumption that one can passively “observe” a disease-relevant marker neglects these costs: true system interrogation requires energy, integration, and an acknowledgment of information bottlenecks within noisy, adaptive biological systems [50,51].

Furthermore, Dirac’s Principle, that theoretical frameworks must not oversimplify the realities they seek to describe, warns against the persistent reductionism dominating biomarker discovery. Over-simplified, linear models fail to capture the inherently dynamic, nonlinear, and non-equilibrium nature of living systems. Consequently, much of biomarker research is trapped by a naive premise: that complex biological states can be linearly decomposed into discrete parts, rather than arising from integrated network interactions [52].

Taken together, these perspectives argue that current biomarker strategies are not merely limited by technological sophistication, but by deeper conceptual misunderstandings of biological complexity. To transcend these limitations, we must shift towards system-level measurements, information-aware modelling, and dynamic, integrative biomarkers.

In this context, 3D genomic profiling offers a transformative opportunity. By reflecting the spatial organization of the genome, itself shaped by the cumulative pressures of the cellular microenvironment, systemic physiological dynamics, and exposure to external environmental signals, the 3D genome acts as a true integrator of informational events [13]. It captures how living systems reconcile intrinsic stochasticity and external chaos into coherent regulatory programmes that underpin cell fate decisions and disease phenotypes. Rather than relying on static snapshots of molecular abundance, the architecture of the 3D genome encodes the history, current state, and adaptive potential of a cell within its chromatin landscape. As such, 3D genomics holds the promise of providing biomarkers that are not merely predictive, but dynamically reflective, of system-level biology, finally offering a bridge between measurement, computation, and the emergent intelligence of living systems.

This schematic illustrates how foundational limitations from complex systems theory constrain current approaches to biomarker discovery in molecular biology. The top row highlights three conceptual challenges: (1) the El Farol Bar paradox illustrates unpredictability due to decision-making based on incomplete information; (2) Maxwell’s Demon underscores the energetic and physical costs of acquiring perfect information; and (3) Dirac’s Principle warns against oversimplified models that fail to capture biological complexity. Together, these concepts point to the inadequacy of fragmented molecular measurements—such as isolated biomarkers or reductionist models—which contribute to irreproducibility and poor clinical predictive performance. In contrast, 3D genomic profiling captures the spatial organization of the genome and encodes cellular, systemic, and environmental pressures, acting as an integrative readout of a dynamic system state. By bridging stochasticity and predictability, 3D chromatin architecture offers a more holistic, systems-level solution for biomarker discovery.

## 4. Conclusions

Three-dimensional genomic profiling of blood, using the EpiSwitch platform, together with AI-based EpiSwitch Data Knowledge analysis, has revealed statistically significant and conditional long-range chromosomal interactions in patients with ulcerative colitis (UC), amyotrophic lateral sclerosis (ALS), systemic lupus erythematosus (SLE) and primary progressive multiple sclerosis (MS-PP), as well as responders and non-responders to immune checkpoint inhibitors (PD-1). Each condition is associated with *specific clusters* of 3D interactions across the genome, including interactions found in the vicinity of the auto-immunity risk locus *IL2RA*, illustrated in Figure 5. Autoimmune conditions such as these are often characterized by the predominance of pro-inflammatory T helper (Th17) cells over regulatory T cells (Tregs).

An independent study in search of the mechanisms that control polarization of naïve T cells analysed functional regulatory elements around *IL2RA* in Jurkat cells, using CRISPR activation and tiling over a 178 kb region with 20,000 sgRNAs [14]. By integrating CRISPRa results with chromatin accessibility, histone modifications and HiChIP, the study identified several regulatory regions, including an element that harbours an autoimmunity risk variant rs61839660 for Crohn’s disease and type 1 diabetes (T1D). Using a mouse model further demonstrated that perturbation of the underlying sequence, annotated as CaRE4, did not affect the level of expression of *IRL2A* per se, but the *timing* of the activation. A dynamic short delay in activation skewed polarization of naïve T cells towards IL-17 secreting pro-inflammatory T helper (Th17) cells and away from a regulatory T state (Tregs), one of the hallmarks of Crohn’s disease [14] and other autoimmune conditions. Comparison of this ex vivo and animal model data with clinical EpiSwitch 3D genomic profiles obtained from patients with ulcerative colitis (UC), amyotrophic lateral sclerosis (ALS), systemic lupus erythematosus (SLE) and primary progressive multiple sclerosis (MS-PP) reveals that the same CaRE4 region is involved (Figure 5).

CaRE4 encompasses an EpiSwitch chromosomal loop anchor in these patients, one part of a conditional and statistically significant 3D genomic biomarker, with the second anchor located in the neighbouring *RBM17* gene. Remarkably, the *RBM17* located loop anchor is associated with instructive interactions involving other loop anchors within and upstream of *IL2RA,* specific for Lupus Erythematosus Systemic (SLE), Rheumatoid Arthritis (MTX), Relapsing Remitting Multiple Sclerosis (MSRR) and response to immune checkpoint inhibitor treatment (PD-1) (Figure 5). Further analysis using AI-based EpiSwitch Data Knowledge revealed that the conditional 3D genome architecture in this region is associated with an even bigger panel of autoimmune conditions, including Allergic Reaction, Ankylosing Spondylitis, Asthma, Autoimmune Disease Multisystem Infantile-Onset 1, Autoimmune Disease Multisystem Infantile-Onset 2, Autoimmune Thyroiditis, Celiac Disease, Cholangitis Sclerosing, Common Variable Immunodeficiency, Diabetes Mellitus Insulin-Dependent, Eczema, Juvenile arthritis and Psoriasis, suggesting there is much more to discover about the molecular basis of these conditions. Importantly, the AI-based EpiSwitch Data Knowledge, combined with EpiSwitch analysis of 3D interactions in patient blood, provides important clues as to where experimental models should be targeted, meaning that expensive and technically challenging screens can be focused on clinically validated targets in their scope. Further understanding of the relationship between 3D genomic dynamics and mechanisms of epigenetic and gene expression controls is one of the biggest challenges in current research into genome architecture.

When it comes to direct use of 3D genomics in disease biomarker development and clinical testing, one can compare the earlier mentioned blood-based PSE test for detection of prostate cancer [3] with the broad variety of alternative clinical diagnostic tests [53], employing protein, mRNA and cell-free DNA readouts in blood or urine (Table 1). The efficacy of stratification by PSE, a reflex test based on five systemic 3D genomic biomarkers and the standard PSA readout, is markedly higher and more balanced across all the confusion matrix parameters. It is particularly noticeable when comparing different molecular approaches for their positive predictive values (PPVs)—a measure of validity for the positive outcome of the individual test for the potential patient—with PSA (protein-based) at 25%, Stockholm (complex protein- and genetic-based) at 53%, and MyProstateScore 2.0 (complex transcript-based) at 28% [53,54], compared to PSE at 93%.

Altogether, 3D regulatory profiling challenges the prevailing molecular paradigm that traditionally reduces disease mechanisms to static sequence variation and isolated gene-expression events. Instead, it highlights the importance of chromatin architecture as a biochemical manifestation of cross-genome conditional interactions and as a network integrator of dynamic cellular, systemic, and environmental cues [13]. Subtle temporal shifts in the activities of regulatory elements linked to spatial genome organization can have pronounced effects on gene regulation, immune balance, and disease susceptibility, effects that could easily evade detection by a conventional linear steady-state assay.

Moving forward, a more complete understanding of human biology will require models that account for the regulatory three-dimensional structure of the genome, its logic, and the flow of the information to and from the 3D genomic architecture in a living cell. Integrating these principles into biomarker discovery and therapeutic design, along with other frontline approaches in precision medicine focused on multi-omics and other system-level tools, will be essential for advancing precision medicine beyond its current limitations.

## Figures and Tables

**Figure 1 cancers-17-02187-f001:**
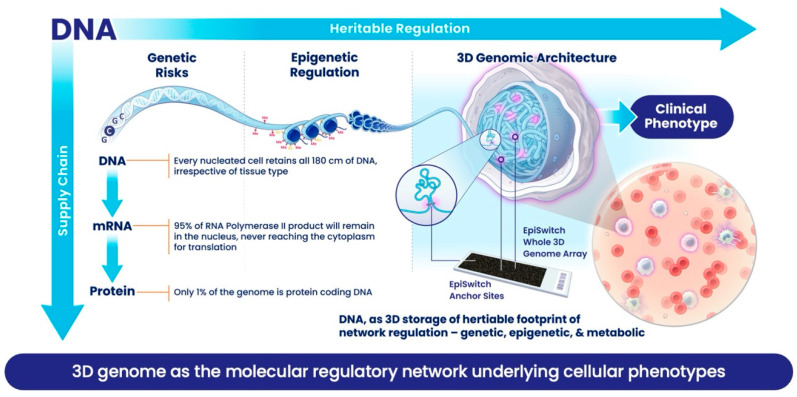
Three-dimensional genomic architecture is a footprint of the cellular epigenetic and metabolic state of the cell. It is a heritable and stable conditional profile which acts as a biomarker for clinical outcomes, based on a readout from white blood cells. Note that RNA and DNA feature properties of the supply chain component and, while supporting the growth of the cell, this does not necessarily reflect the cell’s immediate epigenetic and metabolic state associated with the 3D architecture and phenotype.

**Figure 2 cancers-17-02187-f002:**
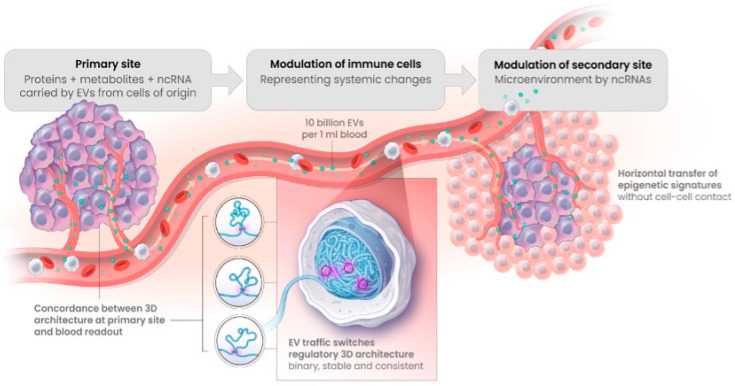
Exosomes as extracellular vesicles (EVs) actively spread epigenetic information from the primary sites of molecular pathology, triggering consistent changes in 3D architecture profiles across many tissues, including WBC.

**Figure 3 cancers-17-02187-f003:**
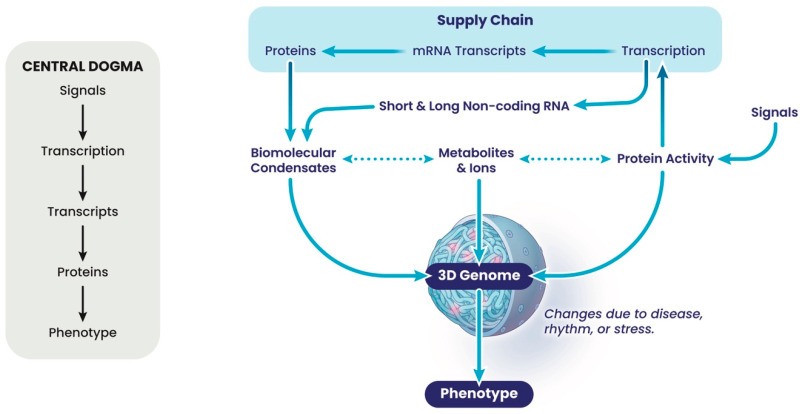
An alternative view of relationships in the cell that contribute to the phenotype. The left panel is a simple representation of the supply chain derived from the Central Dogma. The right panel illustrates how the supply chain (blue rectangle) is integrated with other factors that influence folding of the 3D genome and the cellular phenotype.

**Figure 4 cancers-17-02187-f004:**
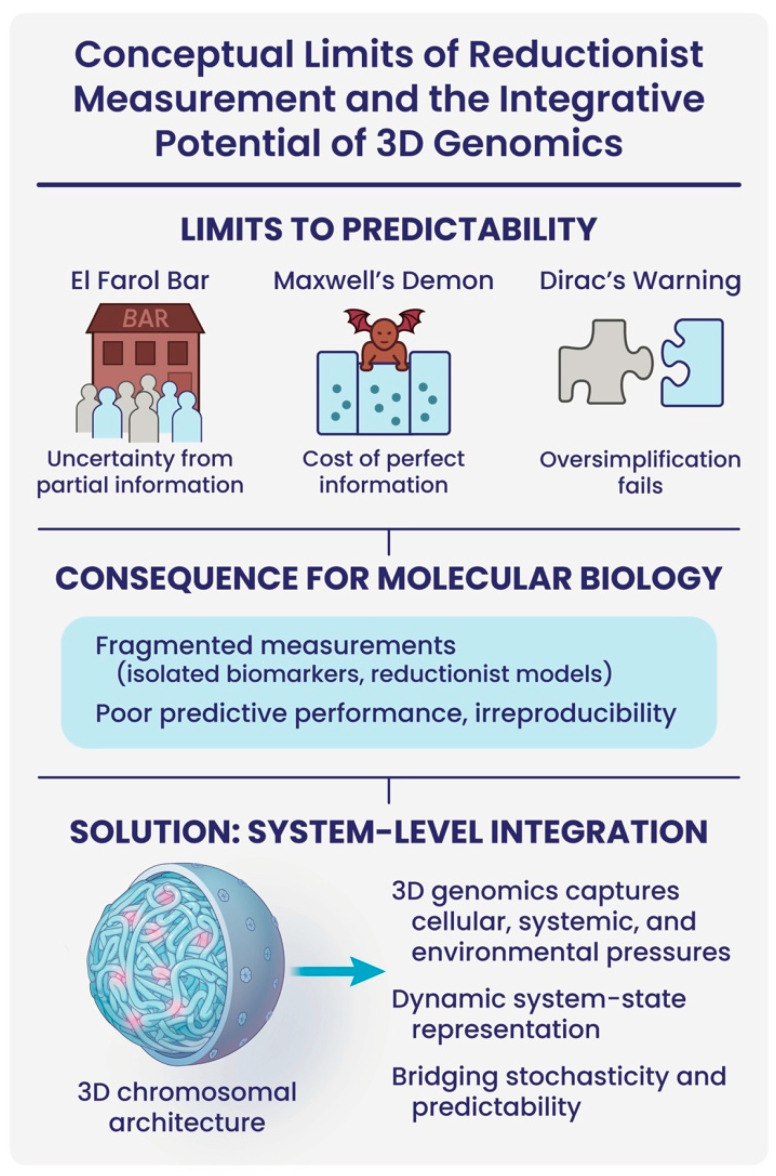
Conceptual limits of reductionist measurement and the integrative potential of 3D genomics.

**Figure 5 cancers-17-02187-f005:**
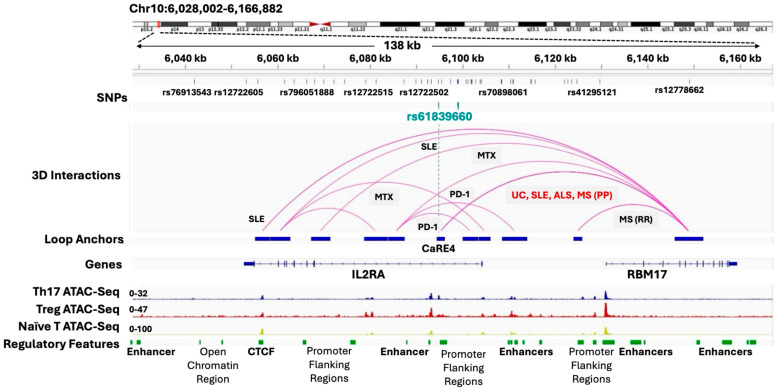
Integrative Genomic Viewer (IGV) image of the *IL2RA* locus combining data from Simeonov et al. [14] with 3D genomic profiling using EpiSwitch clinical testing in blood, together with the AI-based EpiSwitch Data Knowledge. CRISPRa-responsive element 4 (CaRE4), identified in Jurkat cells, overlaps with rs61839660 (cyan) and a key anchoring site (blue) for statistically significant conditional long-range 3D chromosome interaction (purple) specific for auto- and neuro-immune conditions UC, SLE, ALS and MS(PP). Regions of open chromatin (ATAC-Seq) and associated regulatory features are indicated below the genes. Clinical conditions associated with specific 3D genomic interactions: MS(MR)—multiple sclerosis relapsing remitting, MS(PP)—multiple sclerosis primary progressive, UC—ulcerative colitis, SLE—systemic lupus erythematosus, ALS—amyotrophic lateral sclerosis (as neuroinflammation), MTX—response to methotrexate in rheumatoid arthritis, PD-1—response to immune checkpoint inhibitor of PD-1 in oncological conditions. Note that the top *IL2RA* SNPs are associated with Allergic Reaction, Ankylosing Spondylitis, Asthma, Autoimmune Disease Multisystem Infantile-Onset 1, Autoimmune Disease Multisystem Infantile-Onset 2, Autoimmune Thyroiditis, Celiac Disease, Cholangitis Sclerosing, Common Variable Immunodeficiency, Crohn Disease, Diabetes Mellitus Insulin-Dependent, Eczema, Juvenile arthritis, Lupus Erythematosus Systemic, Multiple Sclerosis, Psoriasis, Rheumatoid Arthritis, Ulcerative Colitis.

**Table 1 cancers-17-02187-t001:** Comparative overview of prostate-cancer-detection biomarker modalities, with performance metrics [3,53,54].

Prostate Cancer Test		Performance Metrics			Modality	
Accuracy		PPV ^‡^	NPV ^‡^	Specificity	Sensitivity	
*Next-gen liquid biopsies*						
**EpiSwitch PSE**	**94%**	**93%**	**95%**	**97%**	**86%**	Blood—five EpiSwitch 3D genomic markers, PSA
Prostate health index, PHI	63%	59%	88%	53%	90%	Blood—based on PSA (combines total, free, [-2]pro PSA)
Stockholm3	59%	53%	84%	33%	92%	Blood—protein and genetic markers, clinical information
Select MDx	57%	42%	82%	50%	77%	Urine—mRNA biomarkers (requires digital rectal exam)
IsoPSA	55%	69%	72%	42%	89%	Blood—PSA protein structure for csPCA
MyProstateScore 2.0	50%	28%	97%	32%	96%	Urine—18 gene transcripts (requires digital rectal exam)
ExoDx	50%	36%	91%	34%	92%	Urine—exosome-based genomic test
Proclarix	40%	47%	80%	19%	94%	Blood—protein markers
4Kscore	38%	34%	96%	27%	97%	Blood—four PSA and hk2 markers, clinical information
Galleri	—	—	—	—	11%	Blood—circulating cell-free DNA
*Standard tests*						
Prostate biopsy	89%	90%	74%	96%	48%	Invasive pathology
Prostate-specific antigen, PSA	55%	25%	86%	53%	64%	Blood—protein marker
MRI imaging	48%	51%	89%	41%	93%	Imaging

^‡^ PPV: positive predictive value; NPV: negative predictive value.

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
