# Peer review of "Paradigm Lost"

_cancers, 2025, doi:10.3390/cancers17132187_

Round 1
Reviewer 1 Report
Comments and Suggestions for Authors
The manuscript titled "Paradigm Lost" by Jane Mellor, Ewan Hunter, and Alexandre Akoulitchev presents an intellectually stimulating critique of the central dogma of molecular biology, advocating for a paradigm shift toward the 3D genomic architecture as a superior biomarker framework for precision medicine. This article opens new conceptual terrain, suggesting an emergent model of biomolecular regulation which is not structured around a linear "supply chain" of information flow, but around the dynamic, spatially organized regulatory networks encoded in chromatin architecture.
The extensive use of metaphors and analogies throughout the manuscript is conceptually stimulating and makes the article both intellectually provocative and a genuinely engaging read. That said, I offer a few suggestions that may further strengthen this already compelling and well-crafted perspective:
- While 3D genomics is presented as the inevitable next step, the manuscript the article would benefit from a deeper engagement with other cutting-edge approaches in precision medicine, particularly multiomics, spatial transcriptomics, and high-resolution single-cell profiling. Rather than positioning 3D genomics as a replacement for these systems-level tools, the article could highlight its potential to synergize with them, offering a more integrative vision of the future.
- The authors briefly allude to techniques such as 3C-based chromosome conformation capture and AI-assisted data analysis. However, given the centrality of these methods to the article's core argument, a more detailed overview would be beneficial. Specifically, readers would benefit from a clearer description of the experimental workflow: how chromatin interactions are captured, processed, and validated, as well as the computational pipeline for feature selection, model training, and biomarker interpretation. A concise overview of these steps would help contextualize the robustness and reproducibility of the approach being advocated.
- A concise table comparing 3D genomic biomarkers with traditional central dogma-derived biomarkers across areas like accuracy, stability, scalability, and biological insight would help crystallize the manuscript’s arguments and aid broader comprehension.
- A forward-looking discussion would further enhance the manuscript. Key questions might include: What are the technological and biological limitations currently facing 3D genomics? What infrastructural, regulatory, and economic developments are required to facilitate the integration of 3D genomic profiling into routine clinical practice?
Reviewer 2 Report
Comments and Suggestions for Authors
This manuscript presents a compelling and conceptually rich critique of traditional reductionist paradigms in biomedical research. The authors argue that linear models based on the Central Dogma (DNA → RNA → Protein) are insufficient to capture the complexity, dynamism, and nonlinearity inherent in biological systems. In response, they advocate for a systems biology approach that incorporates 3D genomic architecture, epigenetic modifications, and integrative, network-based methodologies. The discussion is intellectually engaging, with interdisciplinary references (e.g., Maxwell’s Demon, Dirac’s Principle) that broaden the scope of the argument. The manuscript aligns well with emerging trends in precision medicine and highlights recent advances such as chromatin conformation capture and machine learning as tools to support a more systemic understanding of health and disease. Overall, it contributes meaningfully to ongoing discussions on paradigm shifts in biomedical science and is positioned as a thought leadership piece.
However, several areas could be strengthened to enhance clarity, depth, and practical applicability.
-
The manuscript would benefit from the inclusion of concrete case studies or real-world examples demonstrating successful implementation of systems biology approaches in clinical diagnostics or therapeutic development. Citing specific studies where integrative models have outperformed reductionist methods in disease stratification, biomarker discovery, or treatment response prediction would lend practical weight to the arguments.
-
While the manuscript discusses chromatin architecture in detail, it underrepresents other important regulatory mechanisms. Epigenetic features such as DNA methylation and histone modifications, which play essential roles in transcriptional regulation and cellular identity, are mentioned only briefly. A more integrated discussion of how these interact with 3D genome organization would provide a more complete picture of gene regulation dynamics.
-
The role of non-coding RNAs—such as microRNAs, long non-coding RNAs (lncRNAs), and circular RNAs—in post-transcriptional regulation is largely absent. These molecules have well-established functions in modulating gene expression and are increasingly recognized as biomarkers and therapeutic targets. Their inclusion would deepen the discussion on regulatory complexity.
-
Cellular heterogeneity and the advent of single-cell technologies are central to current systems biology efforts, especially in understanding tumor microenvironments, immune responses, and tissue regeneration. Surprisingly, these aspects are not prominently featured in the manuscript. A discussion of how single-cell and spatial omics approaches are reshaping our understanding of disease would significantly strengthen the paper.
-
The manuscript touches on technological innovation but could benefit from a more balanced and detailed discussion of current limitations in the field. Issues such as data heterogeneity, lack of standardization, computational burden, and challenges in integrating multi-layered omics data are critical to both the adoption and credibility of systems approaches. Addressing these challenges, and potentially outlining future directions or solutions, would enhance the manuscript’s utility and realism.
-
While the manuscript is conceptually rich, certain sections are densely written and may be difficult for readers who are less familiar with the theoretical frameworks employed. Simplifying or summarizing complex ideas, perhaps through the use of illustrative figures or summary tables, could improve clarity without sacrificing intellectual depth.
-
The manuscript critiques reductionist paradigms effectively but might overstate their limitations without fully acknowledging the progress made in integrating multi-omics data and building hybrid models that combine reductionist and systems-level insights. A more nuanced discussion recognizing these efforts would foster a more balanced and constructive tone.
-
The translational potential of the proposed paradigm shift is not fully explored. For example, how might systems biology approaches be implemented in routine clinical workflows? What are the regulatory, infrastructural, or economic barriers to adoption? Including such considerations would bridge the gap between theory and practice and make the manuscript more actionable for a wider audience.
-
The manuscript briefly references machine learning and artificial intelligence but could elaborate more on how these tools are being used in predictive modeling, especially in the context of single-cell multi-omics, spatial transcriptomics, and dynamic network inference. Highlighting current successes and open challenges in this area would complement the conceptual arguments with technological relevance.
Overall, while the manuscript offers a timely and intellectually stimulating vision for the future of biomedical research, incorporating the above points would help ensure that its theoretical depth is matched by practical relevance and accessibility. In my vision, these revisions would also broaden its appeal to both academic researchers and clinical practitioners interested in systems-level thinking.
